# Iron Deficiency Treatment in Heart Failure—Challenges and Therapeutic Solutions

**DOI:** 10.3390/jcm14092934

**Published:** 2025-04-24

**Authors:** Lucreția Anghel, Ciprian Dinu, Diana Patraș, Anamaria Ciubară, Iulia Chiscop

**Affiliations:** 1Saint Apostle Andrew Emergency County Clinical Hospital, 177 Brailei St., 800578 Galati, Romania; anghel_lucretia@yahoo.com (L.A.); patras.diana@yahoo.com (D.P.); 2Faculty of Medicine and Pharmacy, “Dunărea de Jos” University, 35 AI Cuza St., 800010 Galati, Romania; 3Dentistry Department, Medicine & Pharmacy Faculty, Dunarea de Jos University, 47 Domneasca Str., 800008 Galati, Romania; ciprian.dinu@ugal.ro; 4Doctoral School Biomedicine Science, University Galati, 800008 Galati, Romania; 5Clinical Surgical Department, Faculty of Medicine and Pharmacy, “Dunărea de Jos” University, 800008 Galati, Romania; iulia.chiscop@ugal.ro

**Keywords:** iron deficiency, heart failure, intravenous iron, ferric carboxymaltose, transferrin saturation

## Abstract

Iron deficiency (ID) is a common comorbidity in heart failure (HF), affecting nearly 50% of patients and worsening symptoms, exercise capacity, and prognosis. This review summarizes recent evidence from meta-analyses, clinical trials, and guidelines on the pathophysiology, diagnosis, and treatment of ID in HF. ID in HF results from chronic inflammation, intestinal congestion, and impaired iron metabolism. Diagnosis is based on serum ferritin and transferrin saturation (TSAT) levels. While oral iron therapy has limited efficacy, intravenous iron, particularly ferric carboxymaltose and ferric derisomaltose, improves symptoms and exercise tolerance and reduces hospitalizations. Timely diagnosis and treatment of ID in HF are essential. Intravenous iron is the preferred therapeutic approach, but further research is needed to optimize long-term management.

## 1. Introduction

Heart failure (HF) is a clinical syndrome characterized by typical symptoms (e.g., dyspnea, fatigue) and signs (e.g., edema) caused by structural and/or functional cardiac abnormalities, resulting in reduced cardiac output and/or elevated intracardiac pressures [1]. Despite progress in pharmacological and device-oriented treatments, morbidity and mortality rates persist at elevated levels [2]. Iron deficiency affects approximately 50% of heart failure patients, with prevalence rates ranging from 30 to 70% depending on disease severity and comorbidities, independent of anemia status [3].

Iron is essential for oxygen delivery, mitochondrial activity, and cellular energy generation. A deficit inhibits skeletal muscle and cardiac function, resulting in exercise intolerance and fatigue—characteristic signs of heart failure [4]. Moreover, iron is crucial for erythropoiesis, enzymatic processes, and immunological function, underscoring its significance in sustaining systemic homeostasis [5].

Recent research indicates that correcting iron deficiency in heart failure patients enhances symptoms, exercise capacity, and therapeutic outcomes. Recent clinical trials, including FAIR-HF (Ferinject Assessment in Patients with Iron Deficiency and Chronic Heart Failure) [6], CONFIRM-HF (Ferric Carboxymaltose Evaluation on Performance in Patients with Iron Deficiency in Combination with Chronic Heart Failure) [7], and AFFIRM-AHF (A Study to Compare Ferric Carboxymaltose with Placebo in Patients with Acute Heart Failure and Iron Deficiency) [8], have demonstrated that intravenous (IV) iron supplementation (e.g., ferric carboxymaltose) improves functional capacity and quality of life (QoL) and reduces HF-related hospitalizations. These findings indicate that iron repletion has become a significant therapeutic objective in heart failure care, leading to the incorporation of iron deficiency screening and treatment in worldwide recommendations [9].

Despite its clinical importance, ID is often misdiagnosed and inadequately treated due to inconsistencies in terminology, insufficient knowledge, and difficulties in differentiating between absolute and functional iron insufficiency [10]. Absolute iron deficiency denotes reduced iron reserves, but functional ID arises when iron stores are sufficient, although their mobilization and use are hindered by chronic inflammation and hepcidin overexpression [11]. The intricate relationship between heart failure pathogenesis and iron metabolism requires a comprehensive grasp of diagnostic standards and treatment approaches to enhance patient care.

This review presents a current summary of the etiology, diagnostic criteria, and therapeutic approaches for iron deficiency in heart failure, with a specific focus on the significance of intravenous iron therapy. This study aims to enhance outcomes for heart failure patients with iron insufficiency by addressing present issues and investigating potential future options.

## 2. Materials and Methods

This narrative review was conducted to synthesize current evidence on the pathophysiology, clinical implications, and treatment of iron deficiency in heart failure. A structured literature search was performed using databases including PubMed, Scopus, and Web of Science. Search terms included combinations of the following: “iron deficiency”, “heart failure”, “intravenous iron therapy”, “ferric carboxymaltose”, “functional iron deficiency”, and “clinical outcomes”.

The inclusion criteria were (1) original articles, reviews, meta-analyses, or clinical guidelines published between January 2015 and January 2025; (2) studies involving adult patients with HF and iron deficiency; and (3) studies evaluating diagnostic criteria, clinical outcomes, or treatment strategies. Articles not in English or not focused on human subjects were excluded.

A total of eleven studies were selected based on relevance, design (RCTs, meta-analyses, observational studies), and alignment with the topic. Each study underwent quality appraisal using the SANRA framework. Appendix A Table A1 presents the SANRA-based quality scoring.

Additionally, international guidelines from recognized societies such as the European Society of Cardiology (ESC) and the American College of Cardiology (ACC) were reviewed. The most relevant and up-to-date findings were selected to ensure a comprehensive and evidence-based discussion.

## 3. Pathophysiology of Iron Deficiency in Heart Failure

### 3.1. Absolute vs. Functional Iron Deficiency

Iron deficiency in heart failure is categorized into two primary types:

Absolute iron shortage denotes a genuine reduction of iron reserves resulting from insufficient consumption, persistent blood loss, or malabsorption. In this condition, ferritin levels are reduced (<100 ng/mL), and iron stores in the liver, bone marrow, and spleen are depleted [12]. Absolute iron deficiency is frequently noted in individuals with inadequate nutritional status, recurrent gastrointestinal hemorrhage, or disorders that impair absorption, like celiac disease or chronic gastrointestinal inflammation [13]. Diagnosis requires both low ferritin (<100 ng/mL) and low transferrin saturation (TSAT < 20%), reflecting systemic iron depletion. Functional iron deficiency, in contrast, arises when iron stores are sufficient (ferritin 100–299 ng/mL) but TSAT remains <20% due to inflammation-mediated iron sequestration [14].

Functional iron deficit arises when iron reserves are maintained or increased, although iron is confined inside the reticuloendothelial system, rendering it unavailable for erythropoiesis or cellular metabolism [15]. This syndrome is predominantly influenced by inflammatory mechanisms that disrupt iron transport, making iron unavailable despite normal or high ferritin concentrations [10]. Functional iron deficiency is more prevalent in chronic conditions such as heart failure, where systemic inflammation enhances hepcidin, the principal regulator of iron homeostasis [16]. Hepcidin interacts with ferroportin, the sole identified iron exporter, leading to its breakdown and sequestering iron inside enterocytes and macrophages. Consequently, iron becomes functionally inaccessible to tissues, resulting in a condition of cellular iron deficiency despite normal total body iron levels [17].

Functional ID is crucial in HF patients due to chronic inflammation causing the improper activation of iron regulatory mechanisms. Despite apparently sufficient total body iron, the transport of iron to tissues, particularly the heart and skeletal muscle, is compromised [18]. As a result, individuals manifest symptoms characteristic of iron deficiency, including weariness and diminished exercise capacity, without necessarily displaying anemia.

Precise classification between these two forms of iron insufficiency is essential for effective therapy, as it informs therapeutic choices and dictates the appropriateness of oral vs. intravenous iron supplementation [19]. The diagnostic differentiation emphasizes the inadequacies of depending just on serum ferritin for evaluating iron status in heart failure, highlighting the necessity for a comprehensive examination utilizing both ferritin and transferrin saturation (TSAT) [20].

### 3.2. Inflammatory Pathways and Hepcidin Regulation

A key feature of HF is chronic low-grade inflammation. Pro-inflammatory cytokines, notably interleukin-6 (IL-6), tumor necrosis factor-alpha (TNF-α), and interleukin-1 beta (IL-1β), are elevated and play a crucial role in systemic inflammation. These cytokines function as mediators that promote hepatic production of hepcidin, a 25-amino acid peptide hormone that is the primary regulator of iron homeostasis [17].

Hepcidin regulates iron metabolism by binding to ferroportin, the sole identified cellular iron exporter, which is present on the basolateral surface of enterocytes, macrophages, and hepatocytes. Upon binding, hepcidin facilitates the internalization and degradation of ferroportin, hence inhibiting iron efflux from these cells into the plasma [21]. This leads to iron sequestration in storage sites and decreased dietary iron absorption, resulting in hypoferremia despite normal or high total body iron levels—a characteristic of functional iron shortage [22].

In HF, chronic systemic inflammation sustains increased hepcidin levels over time. This reaction, although beneficial during acute infection or damage, becomes detrimental in chronic illness conditions such as heart failure [23]. Extended hepcidin increase leads to persistent iron limitation and compromised erythropoiesis, despite the lack of overt anemia. This mechanism also restricts iron availability to metabolically active tissues, such as cardiac and skeletal muscle, exacerbating tiredness and diminishing exercise tolerance [24].

Heart failure (HF) establishes a state of chronic low-grade inflammation through multiple interconnected pathways that collectively disrupt iron homeostasis. The inflammatory cascade in HF originates from both cardiac and systemic triggers. Reduced cardiac output leads to tissue hypoperfusion and consequent hypoxia, which activates endothelial cells and promotes the release of damage-associated molecular patterns (DAMPs). These DAMPs stimulate toll-like receptors (TLRs) on immune cells, triggering nuclear factor kappa-B (NF-κB) signaling and subsequent production of pro-inflammatory cytokines, including interleukin-6 (IL-6), tumor necrosis factor-alpha (TNF-α), and interleukin-1 beta (IL-1β) [25].

Furthermore, oxidative stress and endothelial dysfunction in heart failure may enhance inflammatory signaling and intensify hepcidin expression. The persistent interaction between inflammation and iron dysregulation establishes a detrimental loop that sustains iron shortage and its clinical effects [26]. Comprehending this process elucidates the ineffectiveness of oral iron supplementation in heart failure, since iron, despite absorption, is swiftly sequestered and not transported to the required sites [27].

Due to hepcidin’s pivotal function in facilitating inflammation-induced iron sequestration, it is under investigation as a prospective biomarker and therapeutic target. Modulating hepcidin expression or its downstream effects may provide innovative ways to enhance iron homeostasis and clinical outcomes in heart failure patients [28].

### 3.3. Intestinal Congestion and Malabsorption

A major factor in iron shortage in HF is gastrointestinal (GI) congestion and impaired nutritional absorption. In individuals with right-sided or biventricular heart failure, elevated venous pressure results in intestinal wall edema, mucosal ischemia, and diminished gastrointestinal perfusion. These alterations disrupt the normal functioning of enterocytes and hinder nutritional absorption, particularly iron [29].

Iron absorption predominantly takes place in the duodenum and proximal jejunum, contingent upon many factors: an undamaged mucosal surface, acidic stomach pH, and the availability of iron transport proteins [30]. In heart failure, intestinal obstruction impairs this process via many mechanisms:Edema and hypoxia in the intestinal mucosa diminish the absorptive surface area and disrupt enterocyte activity, hence impairing iron absorption.Impaired gastric emptying and intestinal motility may further hinder nutritional transit and absorption efficiency [31].Modifications in the expression of iron transporters, including divalent metal transporter 1 (DMT1) and ferroportin, have been seen in conditions of chronic inflammation and hypoxia, prevalent in progressive heart failure [32].

The regular use of drugs that influence stomach acid production exacerbates this issue. Proton pump inhibitors (PPIs), frequently used for gastroesophageal reflux or as gastroprotection in patients on antiplatelets or NSAIDs, markedly elevate stomach pH, hence decreasing the solubility and consequent absorption of non-heme dietary iron [33].

These gastrointestinal problems generate an unfavorable environment for oral iron absorption, explaining the restricted effectiveness of oral iron supplementation in heart failure patients. Even with sufficient food intake, inadequate absorption leads to both absolute and functional iron insufficiency [34].

This emphasizes the significance of identifying intestinal congestion as a separate and clinically pertinent factor in iron deficiency in heart failure, especially in patients exhibiting symptoms of right-sided heart failure or persistent volume overload [35]. In these instances, intravenous iron treatment is the preferred method of replenishment, circumventing the impaired gastrointestinal system and enhancing iron availability more effectively [36] (Figure 1).

### 3.4. Nutritional Deficiencies

Nutritional deficiencies are often neglected yet significant factors in iron deficiency among heart failure patients. Inadequate consumption of iron-rich foods is prevalent, especially among the elderly, persons with severe illnesses, and those of low socioeconomic position [37]. In heart failure, the burden of chronic disease frequently causes appetite loss, taste alteration, weariness, and sadness, which can lead to reduced calorie and micronutrient consumption [38].

Iron is present in two primary dietary forms: heme iron, sourced from animal products such as red meat, poultry, and fish, and non-heme iron, derived from plant-based sources like legumes, grains, and leafy greens [39]. Heme iron is absorbed more easily by the gastrointestinal system, but non-heme iron needs a conducive stomach environment and the presence of enhancers such as vitamin C for maximum absorption. Heart failure patients frequently ingest fewer animal products due to dietary constraints or preferences, hence further diminishing bioavailable iron intake [40].

Additionally, several food elements frequently ingested by heart failure patients may impede iron absorption. Calcium competes with iron for intestinal absorption. Polyphenols and phytates included in tea, coffee, and whole grains may chelate iron, therefore diminishing its bioavailability. High-fiber diets, frequently advised in heart failure for managing comorbidities, may also hinder iron absorption [41].

Malnutrition constitutes an additional problem. Cardiac cachexia, a severe starvation condition observed in progressive heart failure, is characterized by substantial weight loss, muscular atrophy, and vitamin deficiency, especially iron [42]. In these individuals, iron insufficiency may occur with additional dietary deficiencies, including protein-energy malnutrition, vitamin B12, or folate inadequacy, which can exacerbate the impacts on erythropoiesis and energy metabolism [43].

Given these multiple impacts, dependence on dietary adjustment alone to rectify iron deficiency in heart failure is frequently inadequate. Nutritional advice is crucial for general cardiovascular health; however, it should be supplemented with pharmacological iron therapy—preferably intravenous—for symptomatic individuals with proven insufficiency. Recognizing and mitigating dietary factors related to ID is essential for effective long-term therapy and avoidance of recurrence [44].

### 3.5. Consequences on Cardiac and Skeletal Muscle Function

Iron is crucial for cellular energy metabolism, especially in the mitochondria, where it serves as an important cofactor in the electron transport chain. In HF, ID adversely affects cardiac and skeletal muscle performance by impairing oxidative metabolism, reducing ATP generation, and disrupting mitochondrial function [45].

Cardiac myocytes are cells with a high energy need. Sufficient intracellular iron levels are essential for the optimal operation of mitochondrial respiratory enzymes and iron-sulfur clusters that facilitate ATP production. Iron deficiency results in the downregulation of metabolic pathways, causing diminished energy availability, impaired contractile function, and a decline in cardiac output. This directly leads to heart failure development, as the compromised heart is less capable of fulfilling systemic demands [46].

Skeletal muscle function is similarly impacted. Iron is necessary for myoglobin production and oxidative phosphorylation, both crucial for sustaining muscular endurance and strength. Individuals with ID frequently exhibit symptoms like weariness, diminished activity capacity, and muscular weakness. These signs are not only due to anemia but also indicate a wider systemic effect of iron shortage on muscle energetics [47].

Moreover, iron deficiency has been demonstrated to hinder muscle regeneration and elevate vulnerability to oxidative stress, resulting in muscular atrophy and fatigue. Research utilizing muscle samples and imaging modalities has shown decreased mitochondrial enzyme activity and modified muscle fiber composition in heart failure patients with iron deficiency [48].

Clinical studies, including CONFIRM-HF and EFFECT-HF, have shown that intravenous iron treatment enhances exercise capacity, as seen by six-minute walk distance (6MWD) and peak oxygen consumption (VO2 max), even in non-anemic individuals. This indicates that restoring iron reserves reinstates metabolic function regardless of hemoglobin normalization [49].

Consequently, the ramifications of iron shortage go beyond hematologic metrics and directly influence cardiac and skeletal muscle functionality. Addressing iron deficiency in heart failure is essential not only for rectifying anemia but also for restoring energy deficits and enhancing functional results [50].

## 4. Diagnostic Evaluation of Iron Deficiency in Heart Failure

Precise identification of iron deficiency (ID) in heart failure (HF) is crucial for prompt and suitable intervention. In contrast to the general population, where iron status is often evaluated based purely on hemoglobin or serum ferritin levels, individuals with heart failure need a more sophisticated approach due to the confounding influences of inflammation and comorbidities [20].

### 4.1. Diagnostic Criteria

The existing diagnostic criteria for iron deficiency in heart failure are based on clinical studies and supported by prominent cardiology organizations, including the European Society of Cardiology (ESC). Iron deficiency is characterized by serum ferritin levels below 100 ng/mL, signifying absolute iron deficiency, or serum ferritin levels between 100 and 299 ng/mL accompanied by transferrin saturation (TSAT) below 20%, indicating functional iron insufficiency [51].

This dual-parameter methodology encompasses both absolute and functional iron deficiency and acknowledges that ferritin, an acute-phase reactant, may be high during inflammatory conditions despite diminished iron availability [14]. Ferritin indicates iron storage, whereas TSAT denotes the quantity of circulating iron accessible for erythropoiesis and tissue metabolism. A TSAT under 20% indicates inadequate iron supply to the bone marrow and peripheral tissues, despite ferritin levels being within the normal range [52].

The thresholds were verified in pivotal clinical studies, including FAIR-HF, CONFIRM-HF, and AFFIRM-AHF, which employed these definitions for patient selection and indicated that therapy guided by these criteria results in enhanced clinical outcomes. As a result, these indicators are now incorporated into heart failure care recommendations for regular evaluation [53].

It is essential to recognize that diagnostic cutoffs may vary somewhat in individuals with comorbidities such as chronic renal disease or acute infection, when ferritin levels may be disproportionately increased. In these instances, doctors may need to interpret data judiciously and depend more significantly on TSAT or contemplate the incorporation of other markers [10].

The combination of diminished ferritin and low TSAT provides a pragmatic and clinically significant basis for diagnosing iron insufficiency in heart failure patients and recognizing those who may benefit from iron repletion treatment [54] (Figure 2).

### 4.2. Additional Laboratory Markers

Alongside ferritin and transferrin saturation (TSAT), various additional laboratory markers may aid in diagnosing iron deficiency in heart failure (HF), especially in intricate cases where inflammation, renal impairment, or other comorbidities complicate the interpretation of standard metrics [55].

Serum iron, while often assessed, possesses restricted diagnostic value owing to considerable diurnal fluctuations and susceptibility to recent food consumption. Reduced blood iron concentrations may corroborate the diagnosis of iron insufficiency; nevertheless, they should not be utilized in isolation [56].

Total iron binding capacity (TIBC) indicates the greatest quantity of iron that transferrin in the plasma can bind. An increased TIBC is frequently noted in iron insufficiency; however, similar to serum iron, it may be affected by dietary and inflammatory conditions [57].

The soluble transferrin receptor (sTfR) indicates cellular iron requirement and is less influenced by inflammation compared to ferritin. Increased sTfR levels may signify tissue iron insufficiency despite higher ferritin levels resulting from an acute-phase response. sTfR is very effective in differentiating iron deficiency anemia from anemia of chronic illness; yet, it is not regularly employed in clinical practice due to issues of cost and availability [58].

Reticulocyte hemoglobin content (CHr) offers insight into the functional iron status of erythropoietic cells. A diminished CHr signifies iron-restricted erythropoiesis. It is beneficial for the early identification of iron shortage prior to significant alterations in hemoglobin or ferritin levels; however, its application is predominantly observed in nephrology contexts [59].

Hepcidin, the principal regulator of iron homeostasis, has garnered attention as a diagnostic instrument. Reduced hepcidin levels may indicate absolute iron shortage, but increased levels may imply functional insufficiency resulting from inflammation. Nevertheless, hepcidin assays are not extensively accessible and predominantly reside within the research sector [60].

While ferritin and TSAT are fundamental for detecting iron deficiency in heart failure, these extra biomarkers may offer further insights in particular clinical situations. They can elucidate the diagnosis in patients with equivocal findings or inform therapy recommendations in individuals with many overlapping comorbidities. With advancements in laboratory technologies, certain markers may become increasingly accessible and incorporated into standard practice [14].

### 4.3. Role of Serial Monitoring

Due to the fluctuating characteristics of iron metabolism in heart failure (HF), especially during systemic inflammation, acute decompensation, and the development of comorbidities, continuous assessment of iron parameters is essential for effective therapy. Iron status can vary considerably over time, and singular evaluations may not correctly represent a patient’s continuous requirements or therapy response [61].

Current guidelines advocate for the frequent reassessment of ferritin and transferrin saturation (TSAT), particularly in individuals diagnosed with iron deficiency or those undergoing intravenous iron treatment. In clinically stable individuals, it is prudent to perform iron testing every 3 to 6 months [62]. Increased monitoring may be necessary during periods of clinical decline, following recent hospitalization, or after iron repletion to assess effectiveness and determine the necessity for further treatment [63].

In patients administered intravenous iron, follow-up testing often takes place 4 to 12 weeks after infusion to evaluate the restoration of iron parameters and the amelioration of symptoms. This period permits adequate duration for ferritin levels to stabilize and for TSAT to accurately represent bioavailable iron [64]. An insufficient reaction may suggest continued inflammatory obstruction, sustained hemorrhage, or malabsorption, all of which necessitate additional examination.

Serial monitoring aids in identifying individuals susceptible to iron overload, especially those with chronic renal disease or hepatic failure. While intravenous iron is often safe, excessive or unregulated delivery can lead to increased ferritin levels and oxidative stress [65].

In addition to laboratory indicators, clinical complaints, including weariness, dyspnea, and exercise intolerance, must be routinely reevaluated to maintain consistency between biochemical responses and functional status. Enhancements in 6 min walk test (6MWT) distance or patient-reported quality of life may serve as significant markers of therapeutic efficacy [66].

## 5. Therapeutic Strategies for Iron Deficiency in Heart Failure

The management of ID in HF is to restore iron reserves, alleviate symptoms, promote quality of life, and decrease hospital admissions. In the last ten years, the strategy for iron repletion in heart failure has markedly progressed, with substantial data endorsing intravenous (IV) iron treatment as the recommended method in the majority of instances [67].

### 5.1. Oral Iron Supplementation

Oral iron treatment is extensively utilized in general populations with ID because of its availability, affordability, and non-invasive characteristics. In the setting of HF, its therapeutic value is considerably constrained by physiological and practical obstacles [27].

A significant problem in heart failure is compromised gastrointestinal absorption, frequently caused by intestinal edema from venous congestion, diminished splanchnic blood flow, and altered mucosal function. These variables diminish the efficacy of oral iron absorption in the duodenum and proximal jejunum—essential locations for non-heme iron assimilation [68]. Moreover, numerous heart failure patients are administered proton pump inhibitors (PPIs), which elevate stomach pH and further hinder the solubilization and absorption of oral iron salts [69].

Furthermore, inflammation-induced overexpression of hepcidin—a crucial hormone governing iron metabolism—significantly diminishes iron absorption. Increased hepcidin levels in heart failure block ferroportin, the iron exporter on enterocytes, therefore sequestering iron in intestinal cells and restricting its bioavailability. This tendency is especially significant in functional iron insufficiency, making oral iron supplementation predominantly ineffective in this subgroup [70].

In individuals with extreme iron deficiency, the therapeutic effectiveness of oral iron is frequently undermined by inadequate adherence. Gastrointestinal adverse effects, including nausea, constipation, abdominal cramps, and a metallic taste, are commonly observed and may result in the cessation of medication. The typical therapy duration—generally 8 to 12 weeks—presents a difficulty for patients facing pill load or numerous comorbidities [71].

The IRONOUT-HF trial, a substantial double-blind, placebo-controlled investigation, revealed that oral iron polysaccharide did not substantially enhance exercise capacity or iron indices in patients with heart failure with reduced ejection fraction (HFrEF) during a 16-week duration. These findings have transformed clinical practice to prioritize intravenous iron as the usual treatment [72].

Oral iron may still be deemed appropriate in a few instances: individuals with verified absolute iron deficiency, maintained ejection fraction, mild inflammation, or logistical impediments to intravenous treatment. In these situations, novel formulations with superior tolerability and increased bioavailability—like ferric maltol—may yield better results; but evidence in heart failure groups is still scarce [73].

Due to the intricate pathophysiology of iron metabolism in heart failure and common comorbidities, oral treatment should be limited to meticulously chosen individuals, with vigilant monitoring for both effectiveness and tolerance [29].

### 5.2. Intravenous Iron Therapy

Intravenous (IV) iron therapy has emerged as the primary therapeutic approach for iron deficiency (ID) in patients with heart failure (HF), especially in those with decreased ejection fraction (HFrEF) [74]. In contrast to oral iron, intravenous treatment circumvents the gastrointestinal system, hence evading obstacles such as inflammation, hepcidin-induced iron sequestration, and intestinal malabsorption—prevalent issues in the heart failure population [75].

Numerous randomized controlled studies (RCTs) have conclusively demonstrated the therapeutic advantages of intravenous iron in heart failure (HF). The FAIR-HF study showed substantial enhancements in patient-reported outcomes, New York Heart Association (NYHA) functional classification, and 6 min walk distance (6MWD) subsequent to therapy with ferric carboxymaltose [76,77]. CONFIRM-HF further substantiated these findings, demonstrating enduring enhancements in exercise capacity, quality of life, and decreases in hospitalizations due to deteriorating heart failure throughout a one-year follow-up period [78]. The AFFIRM-AHF study recently demonstrated a decrease in heart failure-related hospitalizations among individuals administered intravenous iron during or immediately following an acute heart failure event [79].

The predominant intravenous iron formulations utilized in heart failure are ferric carboxymaltose and ferric derisomaltose (formerly referred to as iron isomaltoside). These agents possess a good safety profile, exceptional stability, and the capacity to deliver substantial single doses (up to 1000 mg) with low danger of free iron release or hypersensitive reactions [80,81]. This facilitates rapid and effective replenishment of iron reserves in just one or two infusions, enhancing compliance and convenience for both patients and healthcare practitioners [82].

The mechanism of benefit beyond mere treatment of anemia. Intravenous iron treatment replenishes iron levels in peripheral tissues, such as skeletal muscle and the heart, therefore enhancing mitochondrial activity, oxygen transport, and energy generation.

These benefits lead to better exercise tolerance, less tiredness, and higher functional status, even in non-anemic individuals [50].

The European Society of Cardiology (ESC) currently advises the consideration of intravenous iron treatment for symptomatic HFrEF patients exhibiting ferritin levels below 100 ng/mL or ferritin levels between 100 and 299 ng/mL with a transferrin saturation (TSAT) below 20%, supported by substantial clinical data to date [20]. Nonetheless, actual implementation is inadequate, underscoring the necessity for enhanced awareness, infrastructure, and interdisciplinary collaboration to include IV iron into standard heart failure management [83] (Figure 3).

### 5.3. Treatment Protocols and Dosing

The delivery of intravenous (IV) iron in heart failure (HF) must adhere to a systematic approach customized to the patient’s iron status, hemoglobin concentration, body weight, and clinical state. Although particular recommendations may differ depending on the formulation employed, the majority of clinical trials and guidelines advocate for a streamlined and patient-centric dosing approach to guarantee adequate repletion [84].

The typical dose regimen for ferric carboxymaltose is determined by the patient’s weight and hemoglobin levels. Patients often receive 500–1000 mg each infusion during a brief duration (about 15–30 min), with the total replacement dosage determined using simplified dosing tables based on clinical trial procedures. The standard protocol involves administering 1000 mg to individuals over 70 kg and 500–750 mg to those below 70 kg, with further evaluation occurring at 4 to 12 weeks [85].

Ferric derisomaltose, a recognized alternative, facilitates complete iron repletion with a single high-dose infusion of up to 20 mg/kg (often 1000–1500 mg), contingent upon the deficiency’s severity. This single-dose approach provides ease and enhances compliance, particularly for individuals facing logistical or mobility difficulties [86].

It is advisable to monitor serum ferritin and transferrin saturation (TSAT) about 4–12 weeks following the initial dosage to evaluate response and inform further therapy. If iron readings remain abnormal or symptoms continue, a further dosage may be warranted. In patients with persistent risk factors for iron depletion (e.g., gastrointestinal hemorrhage, renal impairment, or chronic inflammation), regular reassessment and maintenance therapy may be required [87].

Avoiding excessive iron repletion is crucial, since it may elevate the risk of oxidative stress and tissue damage. Treatment objectives typically seek a ferritin concentration of 100–300 ng/mL and a transferrin saturation (TSAT) of 20–50%, ensuring a balance between efficacy and safety [88].

In clinical practice, collaboration across cardiology, pharmacy, and infusion services is crucial for ensuring accurate dose, safe administration, and prompt follow-up. A unified treatment regimen in heart failure care pathways might enhance consistency and guarantee that eligible patients obtain the whole advantages of intravenous iron therapy [89] (Table 1).

## 6. Challenges and Future Directions

Notwithstanding persuasive data and guideline recommendations, the therapy of ID in HF has several obstacles that hinder widespread adoption and uniform execution. These difficulties encompass clinical, systemic, and scientific areas, highlighting the necessity for diverse solutions and ongoing study.

### 6.1. Underdiagnosis and Low Awareness

ID is frequently underdiagnosed in HF due to insufficient understanding among healthcare practitioners concerning its incidence, clinical significance, and the differentiation between absolute and functional insufficiency. Numerous doctors persist in linking iron treatment just to anemia, neglecting the wider metabolic and functional ramifications of non-anemic iron deficiency [16].

A major contributing element is the absence of specialized teaching on the topic throughout medical training and ongoing professional development. In contrast to well-established components of heart failure care, such as beta-blocker or diuretic medication, iron deficiency frequently receives insufficient attention in academic curricula and clinical recommendations beyond cardiology-specific contexts [99]. Moreover, the fallacy that normal hemoglobin levels exclude the possibility of iron shortage endures, despite substantial evidence demonstrating that iron deficiency can adversely affect exercise tolerance and quality of life independent of anemia [100].

Clinical inertia is also a contributing factor. In high-traffic clinical settings, practitioners may focus on addressing overt symptoms and comorbidities, so undervaluing the therapeutic importance of subclinical iron deficiency. The lack of visual indicators or early test abnormalities (such as low hemoglobin) renders iron deficiency an often unrecognized factor in ongoing heart failure symptoms [101].

Moreover, the absence of established institutional processes and electronic health record (EHR) prompts leads to uneven screening. In the absence of integrated clinical decision support technologies, chances to evaluate and manage infectious diseases may be missed during standard patient assessments [102].

Mitigating underdiagnosis necessitates coordinated educational initiatives, incorporation of infectious disease screening into heart failure therapy protocols, and creation of accessible resources to aid doctors in identifying and addressing this modifiable risk factor.

### 6.2. Economic and Logistical Constraints

The expense continues to be a substantial obstacle to the extensive adoption of intravenous (IV) iron treatment in several healthcare systems. Despite the evident therapeutic advantages of IV iron, such as decreased hospitalizations and enhanced quality of life, its initial expense is significantly more than that of oral preparations. In areas with constrained healthcare resources or insufficient reimbursement policies for IV iron, clinicians may be dissuaded from commencing treatment [103].

In addition to direct drug expenses, the infrastructure necessary for administering IV iron—comprising infusion rooms, trained personnel, and monitoring apparatus—can restrict accessibility. This is especially pertinent in rural or impoverished regions where access to outpatient infusion clinics is limited. Patients residing in these areas may have extended journey durations, heightened transportation expenses, or logistical challenges in coordinating time away from employment or caring duties to attend appointments [104].

Moreover, ambiguous or contradictory insurance policies and prior authorization stipulations might impede or obstruct access to therapy. The administrative difficulties imposed on doctors and healthcare systems may diminish the willingness to initiate IV iron treatment, even when it is plainly warranted [105].

From the patient’s viewpoint, apprehensions over cost-sharing, out-of-pocket expenditures, and numerous clinic appointments may inhibit participation in therapy. For patients with fixed incomes or inadequate financial literacy, the perceived or actual expenses of care might serve as a significant barrier.

Mitigating these limits necessitates collaborative efforts from governments, payers, and healthcare institutions. Exhibiting the cost-efficiency of intravenous iron by empirical evidence and health-economic modeling might facilitate the justification for expanded coverage and investment. Furthermore, incorporating iron treatment into comprehensive heart failure care models, community infusion programs, or home administration efforts might enhance accessibility while managing [106].

### 6.3. Need for Long-Term Evidence

Although randomized controlled trials (RCTs) like FAIR-HF, CONFIRM-HF, and AFFIRM-AHF have demonstrated that intravenous (IV) iron therapy results in short- and intermediate-term enhancements in exercise capacity, quality of life, and hospitalization rates, there is still a necessity for comprehensive data on long-term outcomes. Specifically, there is little data about the effects of prolonged iron repletion on mortality, heart failure disease progression, and significant adverse cardiovascular events [107].

The majority of accessible studies exhibit restricted follow-up periods, often spanning from 6 months to 1 year. The sustainability of therapeutic advantages and the ideal re-dosing intervals for prolonged care remain ambiguous. The efficacy and cost-effectiveness of administering IV iron at set intervals, guided by symptoms or iron biomarkers, remains uncertain [108].

Furthermore, there exists a paucity of evidence about the administration of IV iron in populations who are underrepresented in existing studies, such as patients with heart failure with preserved ejection fraction (HFpEF), elderly individuals, those with numerous comorbidities, or individuals from varied racial and cultural origins. Incorporating these populations into clinical research is crucial for guaranteeing the generalizability and equality of treatment guidelines [109].

The long-term safety remains a critical domain for more investigation. Despite the absence of notable safety signals thus far, prolonged follow-up studies are essential to verify the lack of cumulative concerns, such as iron overload, oxidative stress, or changes in immune function resulting from repeated or chronic usage [110].

Subsequent research should assess health economic outcomes, including the effects of IV iron on overall healthcare utilization, medication adherence, patient-reported outcomes, and quality-adjusted life years (QALYs). This data are essential for guiding healthcare policy, reimbursement determinations, and clinical practice protocols [111].

### 6.4. Future Research Directions

As the role of iron deficiency in heart failure becomes increasingly recognized, future research efforts must focus on addressing the remaining knowledge gaps and optimizing therapeutic strategies. Expanding the evidence base is crucial to ensuring more personalized, effective, and equitable care.

One critical area is the inclusion of patients with heart failure with preserved ejection fraction (HFpEF), who are currently underrepresented in trials but comprise a growing portion of the HF population. Understanding the prevalence, pathophysiological role, and treatment effects of iron repletion in HFpEF could help extend the benefits observed in HFrEF to a broader population.

Further studies should explore the comparative effectiveness of different IV iron formulations and dosing strategies, including one-time full repletion vs. staggered or maintenance dosing. Identifying optimal redosing intervals will be key for long-term disease management and cost-efficiency.

Biomarker discovery and validation represent another promising avenue. Novel indicators such as hepcidin, soluble transferrin receptor (sTfR), and reticulocyte hemoglobin content (CHr) may enhance diagnostic accuracy and allow for better monitoring of iron status and treatment response.

The use of real-world data and registries could complement randomized trials by capturing long-term outcomes, safety events, and treatment patterns across diverse populations. Health-economic studies should continue to evaluate the cost-effectiveness of IV iron in various healthcare settings, including bundled care models or home infusion programs.

Incorporating genomic and proteomic tools may also help identify subgroups of HF patients who are more likely to benefit from iron therapy, paving the way for precision medicine approaches.

## 7. Conclusions

Iron deficiency stands as a pivotal yet frequently overlooked comorbidity in heart failure, weaving a complex pathophysiological tapestry that intertwines chronic inflammation, impaired iron metabolism, and systemic consequences. The profound clinical impact of this condition—manifesting through debilitating fatigue; diminished exercise capacity; and increased hospitalizations—demands our unwavering attention. Through this comprehensive review, we have traced the intricate journey from molecular mechanisms to bedside applications, revealing how iron deficiency disrupts not just hematopoiesis but fundamental cellular energetics in both cardiac and skeletal muscle.

The diagnostic landscape presents both challenges and opportunities, where traditional markers like ferritin and transferrin saturation require careful interpretation against the backdrop of heart failure’s inflammatory milieu. Emerging biomarkers whisper promises of greater precision, yet their clinical adoption remains in its infancy. Therapeutic advances, particularly intravenous iron formulations, have illuminated a path forward, demonstrating consistent improvements in functional status and quality of life across multiple rigorous trials. These clinical benefits extend beyond simple anemia correction, speaking to iron’s fundamental role in mitochondrial function and oxygen utilization.

Yet significant barriers shadow these advancements—knowledge gaps among practitioners, fragmented care pathways, and systemic hurdles to treatment access create a chasm between evidence and everyday practice. The elderly, those with preserved ejection fraction, and underserved populations often find themselves on the wrong side of this therapeutic divide. Our gaze must now turn to implementation science, exploring innovative care models that can deliver these treatments equitably while addressing the practical challenges of infusion logistics and long-term monitoring.

The road ahead demands a harmonious convergence of research and clinical innovation. We must pursue longer-term outcome studies that follow patients beyond the horizon of current trials, develop personalized treatment algorithms that account for individual variations in iron metabolism, and create intelligent health systems that embed iron status monitoring into routine heart failure care. Educational initiatives should target both specialists and primary care providers, fostering a deeper understanding of iron deficiency’s multifaceted presentation.

As we stand at this crossroads, the imperative becomes clear: addressing iron deficiency in heart failure is not merely about correcting a laboratory abnormality but about restoring vitality and function to patients whose lives have been constrained by this treatable condition. The challenge before us is to transform growing evidence into meaningful action, ensuring that every heart failure patient receives the comprehensive care they deserve—one that includes vigilant attention to their iron status as a fundamental component of optimal management.

## Figures and Tables

**Figure 1 jcm-14-02934-f001:**
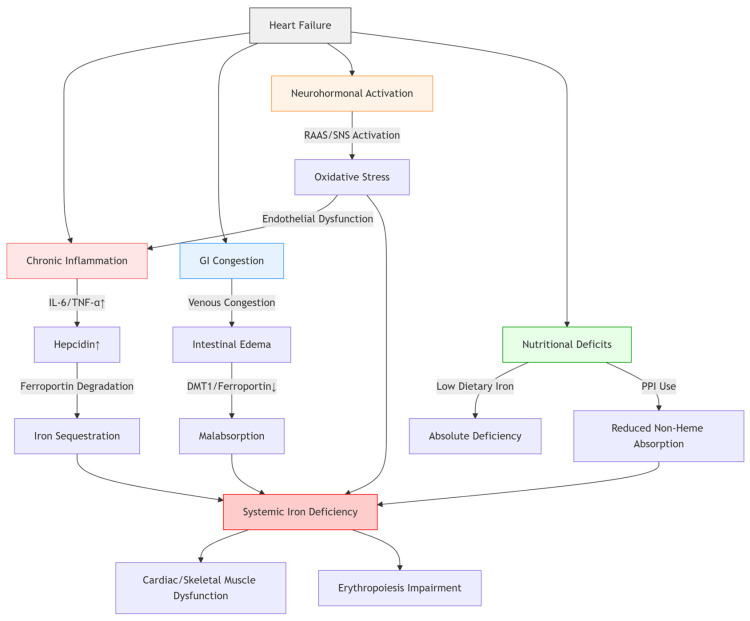
Pathophysiological Mechanisms of Iron Deficiency in Heart Failure.

**Figure 2 jcm-14-02934-f002:**
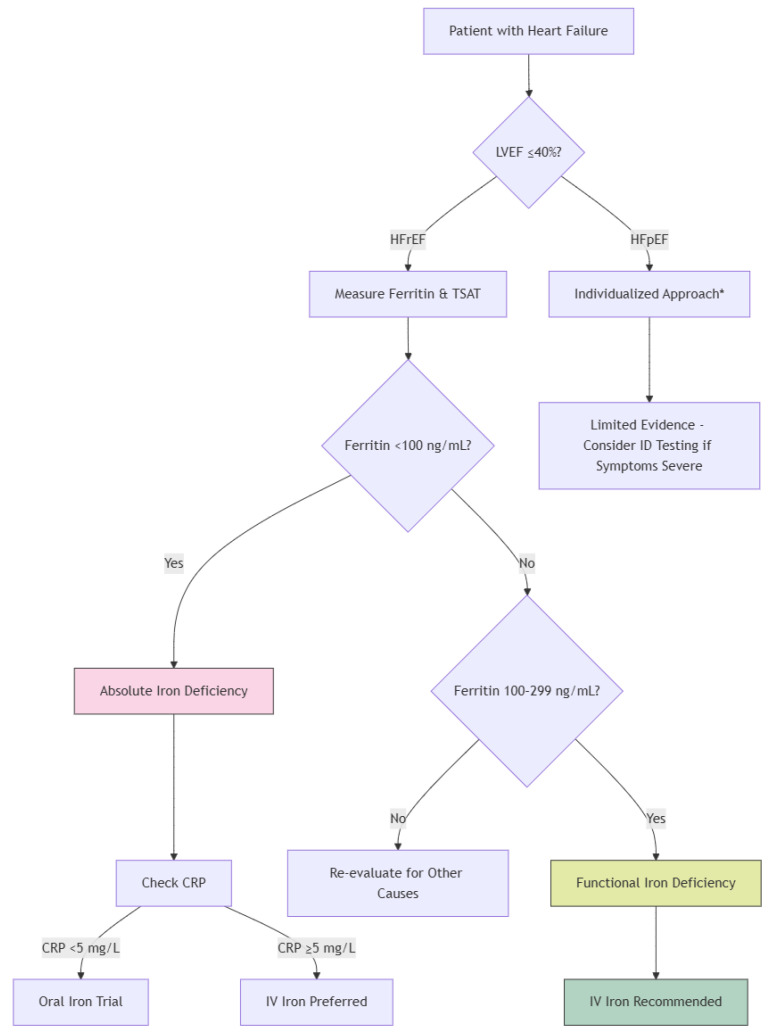
Diagnostic Algorithm for Iron Deficiency in Heart Failure. * Individualized Approach: The decision to proceed with iron testing and supplementation should consider the patient’s overall clinical condition, comorbidities, symptom severity, and other relevant factors. This approach ensures that treatment is tailored to the specific needs and circumstances of each patient.

**Figure 3 jcm-14-02934-f003:**
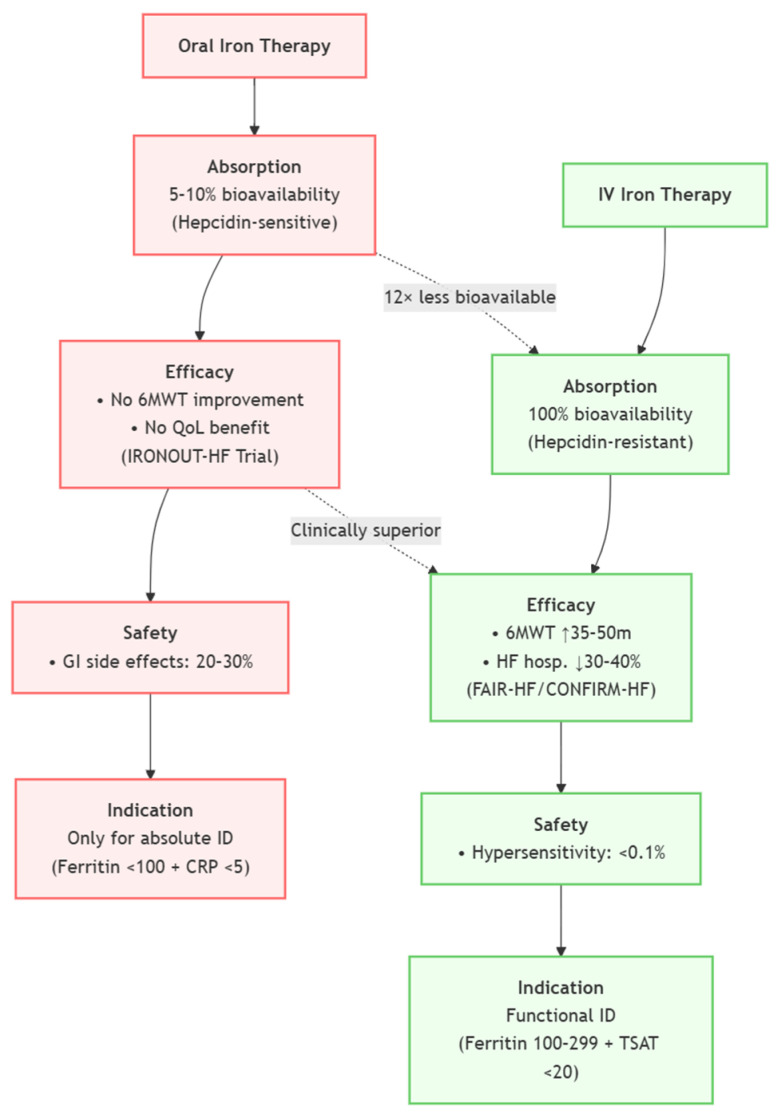
Comparison of Oral vs. Intravenous Iron Therapy in HF (Oral—Left, IV—Right).

**Table 1 jcm-14-02934-t001:** Summary of Clinical Trials and Reviews on Iron Therapy in Heart Failure. (RCT: Randomized controlled trial, FDI: Ferric derisomaltose).

Study	Design	Population	Intervention	Outcomes	Conclusions
Avni et al. [90]	Meta-analysis	HF patients with varying degrees of ID	Pooled IV iron therapy trials	Confirmed ↓ HF hospitalization and ↑ functional status	Meta-analysis supports consistent benefits of IV iron
McDonagh et al. [88]	Prospective multicenter	Real-world HF patients receiving IV iron	Ferric carboxymaltose in guideline-directed care	Validated real-world efficacy of IV iron	Supports implementation of IV iron in practice
Kalra et al. [91]	RCT	HFrEF with iron deficiency, long-term follow-up	Ferric derisomaltose vs. standard care	↓ CV death and HF hospitalization over median follow-up	Ferric derisomaltose reduces long-term HF events
Karavidas et al. [92]	Control trial	HFrEF + ID	Sucrosomial oral iron	↑ Exercise capacity	Sucrosomial effective
Suryani et al. [93]	RCT	HF + ID anemia	Oral ferrous sulphate	↑ Function, correct anemia	Ferrous may help anemia
Sze et al. [94]	Prospective	HFrEF patients with ID stratified by age (<65, 65–74, ≥75)	IV ferric derisomaltose	Improved QoL and exercise capacity in all age groups; safe in elderly	FDI effective and safe across age groups with HFrEF
Chan et al. [95]	Systematic review and meta-analysis	Patients with CKD and HF receiving iron therapy	Various iron therapies (IV + oral)	↓ CV death and HF hospitalization in CKD + HF; further trials needed	Iron therapy shows promise in HF + CKD but needs more RCTs
Ponikowski et al. [96]	RCT	HFrEF, symptomatic, ID	Ferric carboxymaltose (q4 weeks)	Sustained ↑ in 6MWT, ↓ HF hospitalizations	Supports long-term IV iron in HF
van Veldhuisen et al. [97]	RCT	HFrEF with iron deficiency	Ferric carboxymaltose vs. standard care	↑ VO_2_ max, improved functional capacity	Benefits on exercise performance
Ponikowski et al. [98]	RCT	Post-acute HF, LVEF < 50%, ID	Ferric carboxymaltose during/after hospitalization	↓ HF hospitalizations (no mortality effect)	IV iron post-discharge ↓ rehospitalization
Lewis et al. [72]	Rct	HF with reduced EF, ferritin 50–100 or TSAT < 20%	Oral iron (iron polysaccharide) vs. placebo	No significant benefit from oral iron	Oral iron not effective for HF with ID

## Data Availability

Data are contained within the article.

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
