# Peer review of "Iron Deficiency Treatment in Heart Failure—Challenges and Therapeutic Solutions"

_jcm, 2025, doi:10.3390/jcm14092934_

Round 1
Reviewer 1 Report
Comments and Suggestions for Authors
In this review article, the authors provide a comprehensive overview of this emerging comorbidity in patients with heart failure. The discussion on pathophysiology is accurate, the section on diagnosis is also appropriate, and the various therapeutic strategies are well referenced. The final sections addressing current challenges may be particularly well received by readers.
However, we would like to offer several comments for consideration.
1) The review is extremely long, and some concepts are repeated. We would appreciate it if the authors could consider reducing the length of the text.
2) It is important to include the specific data on the high prevalence of iron deficiency, as it is mentioned in the abstract but not in the introduction.
3) In the introduction section: “absolute vs functional iron deficiency,” the discussion of oral iron (lines 92–93) is somewhat confusing. This section should not include treatment-related concepts, as it may mislead the reader. We also suggest reviwing lines 114–116, as they appear to be inconsistent with the information presented later in the treatment section.
4) Figure 2 should be reconsidered, as earlier in the article there is considerable discussion about the recommendation of oral iron, yet this figure presents a different perspective. Additionally, there is no mention of left ventricular ejection fraction, which is an important parameter in heart failure recomendations.
5) Figure 3 should also be reconsidered, as its design is unclear. For example, it is not evident what "low" and "high" are referring to. We kindly suggest reviewing and clarifying this figure.
Author Response
We thank the reviewer for the thoughtful comments and valuable suggestions, which have greatly helped us improve our manuscript. Below are our detailed responses to each point raised:
1) The review is extremely long, and some concepts are repeated. We would appreciate it if the authors could consider reducing the length of the text.
Answer: Thank you for your suggestion; we modified the article according to it.
2) It is important to include the specific data on the high prevalence of iron deficiency, as it is mentioned in the abstract but not in the introduction.
Answer: We thank the reviewer for this insightful observation. We have now incorporated specific prevalence data directly into the introduction.
Iron deficiency affects approximately 50% of heart failure patients, with prevalence rates ranging from 30–70% depending on disease severity and comorbidities, independent of anemia status [2].
3) In the introduction section: “absolute vs functional iron deficiency,” the discussion of oral iron (lines 92–93) is somewhat confusing. This section should not include treatment-related concepts, as it may mislead the reader. We also suggest reviwing lines 114–116, as they appear to be inconsistent with the information presented later in the treatment section.
Answer: We modified the 90-93 lines and removed the 114-116 lines:
Diagnosis requires both low ferritin (<100 ng/mL) and low transferrin saturation (TSAT <20%), reflecting systemic iron depletion. Functional iron deficiency, in contrast, arises when iron stores are sufficient (ferritin 100–299 ng/mL) but TSAT remains <20% due to inflammation-mediated iron sequestration [12].
4) Figure 2 should be reconsidered, as earlier in the article there is considerable discussion about the recommendation of oral iron, yet this figure presents a different perspective. Additionally, there is no mention of left ventricular ejection fraction, which is an important parameter in heart failure recomendations.
Answer: We thank the reviewer for highlighting the inconsistencies related to Figure 2, we modified it.
5) Figure 3 should also be reconsidered, as its design is unclear. For example, it is not evident what "low" and "high" are referring to. We kindly suggest reviewing and clarifying this figure.
Answer: We modified Figure 3.
Reviewer 2 Report
Comments and Suggestions for Authors
Lucrezia Anghel et al. in their comprehensive review aim to evaluate the recent evidence from meta-analyses, clinical studies and guidelines on the pathophysiology, diagnosis and treatment of ID in heart failure. I believe that the authors should change some parts of the review:1. Introduction: define heart failure using the guidelines. Abbreviations of the studies are not reported in the text.2. Materials and methods are accepted.3. The pathophysiology of iron deficiency is well described, however figure 1 is of low quality and does not describe all the mechanisms described in the text. Please describe all the mechanisms described in the text in order to make figure 1 a graphical abstract.4. Diagnostic criteria are well described.5. Therapeutic strategies are well presented, figure 2 is also well represented.6. Table 1 does not present abbreviations, it is also useful to include the results of the different studies cited.7. Challenges and future directions: Long and redundant paragraph. Please reduce unnecessary sentences to make it easier to read.8. The "discussion" paragraph should be "conclusion" and structured like a conclusion.Throughout the manuscript, abbreviations are incorrect (e.g. heart failure...).9. Eliminate conclusions for each subparagraph. It is helpful to include all conclusions in a single final paragraph of the manuscript.10. In paragraph 3.2, describe how heart failure can cause inflammation (10.3390/antiox13070806). I look forward to rereading your manuscript after adding my suggestions.
Author Response
We thank the reviewer for the thoughtful comments and valuable suggestions, which have greatly helped us improve our manuscript. Below are our detailed responses to each point raised:
- Introduction: define heart failure using the guidelines. Abbreviations of the studies are not reported in the text
Answer: We added the paragraph, and added the abreviations:
Heart failure (HF) is a clinical syndrome characterized by typical symptoms (e.g., dyspnea, fatigue) and signs (e.g., edema) caused by structural and/or functional cardiac abnormalities, resulting in reduced cardiac output and/or elevated intracardiac pressures [1].
Recent research indicates that correcting iron deficiency in heart failure patients enhances symptoms, exercise capacity, and therapeutic outcomes. Recent clinical trials, including: FAIR-HF (Ferinject Assessment in Patients with Iron Deficiency and Chronic Heart Failure) [6], CONFIRM-HF (Ferric Carboxymaltose Evaluation on Performance in Patients with Iron Deficiency in Combination with Chronic Heart Failure) [7], and AFFIRM-AHF (A Study to Compare Ferric Carboxymaltose with Placebo in Patients with Acute Heart Failure and Iron Deficiency) [8], have demonstrated that intravenous (IV) iron supplementation (e.g., ferric carboxymaltose) improves functional capacity, quality of life (QoL), and reduces HF-related hospitalizations. These findings indicate that iron repletion has become a significant therapeutic objective in heart failure care, leading to the incorporation of iron deficiency screening and treatment in worldwide recommendations [11].
3. The pathophysiology of iron deficiency is well described, however figure 1 is of low quality and does not describe all the mechanisms described in the text. Please describe all the mechanisms described in the text in order to make figure 1 a graphical abstract
Answer: Thank you for your observation; we modified Figure 1.
6. Table 1 does not present abbreviations, it is also useful to include the results of the different studies cited
Answer: We added the abbreviations:
Table 1. Summary of Clinical Trials and Reviews on Iron Therapy in Heart Failure. (RCT: Randomized controlled trial, FDI: Ferric derisomaltose)
7. Challenges and future directions: Long and redundant paragraph. Please reduce unnecessary sentences to make it easier to read
Answer: We moddified this section.
8. The "discussion" paragraph should be "conclusion" and structured like a conclusion.Throughout the manuscript, abbreviations are incorrect (e.g. heart failure...).
Answer: Thank you for your suggestion, we modified tgis section:
7.Conclusions
Iron deficiency stands as a pivotal yet frequently overlooked comorbidity in heart failure, weaving a complex pathophysiological tapestry that intertwines chronic inflammation, impaired iron metabolism, and systemic consequences. The profound clinical impact of this condition-manifesting through debilitating fatigue, diminished exercise capacity, and increased hospitalizations-demands our unwavering attention. Through this comprehensive review, we have traced the intricate journey from molecular mechanisms to bedside applications, revealing how iron deficiency disrupts not just hematopoiesis but fundamental cellular energetics in both cardiac and skeletal muscle.
The diagnostic landscape presents both challenges and opportunities, where traditional markers like ferritin and transferrin saturation require careful interpretation against the backdrop of heart failure's inflammatory milieu. Emerging biomarkers whisper promises of greater precision, yet their clinical adoption remains in its infancy. Therapeutic advances, particularly intravenous iron formulations, have illuminated a path forward, demonstrating consistent improvements in functional status and quality of life across multiple rigorous trials. These clinical benefits extend beyond simple anemia correction, speaking to iron's fundamental role in mitochondrial function and oxygen utilization.
Yet significant barriers shadow these advancements-knowledge gaps among practitioners, fragmented care pathways, and systemic hurdles to treatment access create a chasm between evidence and everyday practice. The elderly, those with preserved ejection fraction, and underserved populations often find themselves on the wrong side of this therapeutic divide. Our gaze must now turn to implementation science, exploring innovative care models that can deliver these treatments equitably while addressing the practical challenges of infusion logistics and long-term monitoring.
The road ahead demands a harmonious convergence of research and clinical innovation. We must pursue longer-term outcome studies that follow patients beyond the horizon of current trials, develop personalized treatment algorithms that account for individual variations in iron metabolism, and create intelligent health systems that embed iron status monitoring into routine heart failure care. Educational initiatives should target both specialists and primary care providers, fostering a deeper understanding of iron deficiency's multifaceted presentation.
As we stand at this crossroads, the imperative becomes clear: addressing iron deficiency in heart failure is not merely about correcting a laboratory abnormality, but about restoring vitality and function to patients whose lives have been constrained by this treatable condition. The challenge before us is to transform growing evidence into meaningful action, ensuring that every heart failure patient receives the comprehensive care they deserve - one that includes vigilant attention to their iron status as a fundamental component of optimal management.
9. Eliminate conclusions for each subparagraph. It is helpful to include all conclusions in a single final paragraph of the manuscript.
Answer: Thank you, we eliminate them.
10. In paragraph 3.2, describe how heart failure can cause inflammation (10.3390/antiox13070806)
Answer: Thank you for your suggestion, we added the paragraph and also the reference:
Heart failure (HF) establishes a state of chronic low-grade inflammation through multiple interconnected pathways that collectively disrupt iron homeostasis. The inflammatory cascade in HF originates from both cardiac and systemic triggers. Reduced cardiac output leads to tissue hypoperfusion and consequent hypoxia, which activates endothelial cells and promotes the release of damage-associated molecular patterns (DAMPs). These DAMPs stimulate toll-like receptors (TLRs) on immune cells, triggering nuclear factor kappa-B (NF-κB) signaling and subsequent production of pro-inflammatory cytokines, including interleukin-6 (IL-6), tumor necrosis factor-alpha (TNF-α), and interleukin-1 beta (IL-1β) [25].
Round 2
Reviewer 2 Report
Comments and Suggestions for Authors
The authors followed my suggestions for changes to the manuscript. I have no further comments to add.